# Chemical Profile and In Vitro Evaluation of the Antibacterial Activity of *Dioscorea communis* Berry Juice

Konstantina Tsami [1,2,†], Christina Barda [1,3,†], George Ladopoulos [1,2], Nikos Asoutis Didaras [4], Maria-Eleni Grafakou [1,3], Jörg Heilmann [3], Dimitris Mossialos [4], Michail Christou Rallis [2] and Helen Skaltsa [1,*]

1   Department of Pharmacognosy & Chemistry of Natural Products, Faculty of Pharmacy, School of Health Sciences, National & Kapodistrian University of Athens, Zografou, 15771 Athens, Greece; ktsami@pharm.uoa.gr (K.T.); cbarda@pharm.uoa.gr (C.B.); lgeorge4@hotmail.com (G.L.); megrafakou@pharm.uoa.gr (M.-E.G.)
2   Department of Pharmaceutical Technology, Unit of Dermatopharmacology, Faculty of Pharmacy, School of Health Sciences, National & Kapodistrian University of Athens, Zografou, 15771 Athens, Greece; rallis@pharm.uoa.gr
3   Department of Pharmaceutical Biology, Faculty of Pharmacy and Chemistry, University of Regensburg, D-93053 Regensburg, Germany; joerg.heilmann@chemie.uni-regensburg.de
4   Laboratory of Microbial Biotechnology-Molecular Bacteriology-Virology, Department of Biochemistry & Biotechnology, Faculty of Health Sciences, University of Thessaly, 41500 Larissa, Greece; didasout@yahoo.gr (N.A.D.); mosial@bio.uth.gr (D.M.)
*   Correspondence: skaltsa@pharm.uoa.gr
†   These authors contributed equally to this work.

**Abstract:** Within the large family of Dioscoreaceae, *Dioscorea communis* (L.) Caddick & Wilkin (syn. *Tamus communis* L.) is considered among the four most widespread representatives in Europe, and it is commonly known under the name black bryony or bryonia. To date, reports have revealed several chemical components from the leaves and tubers of this plant. Nevertheless, an extensive phytochemical investigation has not been performed on its berry juice. In the present study, metabolite profiling procedures, using LC-MS, GC-MS, and NMR approaches, were applied to investigate the chemical profile of the *D. communis* berries. This work reveals the presence of several metabolites belonging to different phytochemical groups, such as fatty acid esters, alkylamides, phenolic derivatives, and organic acids, with lactic acid being predominant. In parallel, based on orally transmitted traditional uses, the initial extract and selected fractions were tested in vitro for their antibacterial effects and exhibited good activity against two bacterial strains related to skin infections: methicillin-resistant *Staphylococcus aureus* and *Cutibacterium acnes*. The MIC and MBC values of the extract were determined at 1.56% $w/v$ against both bacteria. The results of this study provide important information on the chemical characterization of the *D. communis* berry juice, unveiling the presence of 71 metabolites, which might contribute to and further explain its specific antibacterial activity and its occasional toxicity.

**Keywords:** *Dioscorea communis*; Dioscoreaceae; GC-MC; LC-MS; NMR; antibacterial; *Staphylococcus aureus*; *Cutibacterium acnes*

## 1. Introduction

The genus *Dioscorea* L. is the largest representative of the Dioscoreaceae family, consisting of ≤600 species [1]. The name of the genus, as well as the whole family, was given by the French botanist Charles Plumier in honor of the famous Greek physician Pedanios Dioscorides [2]. *Dioscorea* species are mainly distributed across wet and periodically dry tropical regions, whereas some of them are extended from temperate to alpine climates [3]. In European countries, the Dioscoreaceae family is represented only by four species: *Dioscorea balcanica* Košanin and the formerly known *Borderea pyrenaica* Miégeville

(syn. *Dioscorea pyrenaica* Bubani & Bordère ex Gren), *Borderea chouardii* (Gaussen) Heslot (syn. *Dioscorea chouardii* Gaussen), and *Tamus communis* L. (syn. *D. communis* L.) [4]. The previously recognized genera *Borderea* Miégeville and *Tamus* L. have been integrated into the genus *Dioscorea* due to the high morphological molecular similarities [1,5]. Among them, *Dioscorea* and *Borderea* species are distributed in the Balkan peninsula and the Pyrenees, while *D. communis* is endemic in South, southern Central, and West Europe toward Northern England [4].

　　*D. communis* is a perennial herbaceous climber with large tubers and red berries distributed in the Mediterranean area. Its young stems are part of the traditional diet of many Mediterranean countries, such as Greece, Spain, Portugal, Italy, and Croatia, as a kind of wild asparagus [6]. The plant material is edible only before the flowering period and after cooking, contrary to the completely grown plant considered not edible due to its toxic effects. The species has a great number of synonyms, which is attributed to the high diversity of its phenotype. The most common names are black bryony, bryonia, abronya, or herbe aux femmes battues [6–8]. Different parts of the plant have been used in folk medicine, including the fruits and the roots, which have been applied externally for the treatment of musculoskeletal abnormalities and rheumatism, as well as skin diseases, injuries, and bruises due to their analgesic and rubefacient effect [6,7,9,10]. It is worth mentioning that *D. communis* has been reported (by oral sources) to be used as a traditional remedy in southern Central Greece for skin-related ailments, as well as against oral infections. Previous phytochemical studies on *D. communis* have revealed the presence of saponins [11], phenanthrenes [12], sterols [11], and flavonoid glycosides [13] in the aerial parts and tubers. Regarding bioactivity, *D. communis* has been found to exhibit primarily antioxidant [14,15] and anti-inflammatory effects in vitro and in vivo [16]. Additionally, all parts of the plant are associated with toxicity due to the presence of saponins, calcium oxalate crystals, and histamine. When the berry juice or roots are in contact with skin, they can cause erythematous and papular rash, since calcium oxalate and histamine show high cutaneous penetration and induce allergic reactions [7,17].

　　*Dioscorea* spp., aside from anti-inflammatory, antioxidant [18], antitumor [19], and neuroprotective activity [20], have also shown great antimicrobial activity. Kuete et al. (2012) investigated the antibacterial activity of the methanol extract of the air-dried bulbils of *D. bulbifera* L. var sativa against mycobacteria and multidrug-resistant, Gram-negative bacteria. The results indicated the strong antimicrobial activity of the crude extract against *Escherichia coli*, *Enterobacter aerogenes*, *Klebsiella pneumoniae*, *Mycobacterium smegmatis*, and *M. tuberculosis*, with the MIC rising to 64 µg/mL in all tested bacterial strains [21].

　　Taking these into consideration, the aim of the present study is extensive phytochemical research on the berry juice of *D. communis* using various approaches, including chromatographic methods (CC, LC-MS, and GC-MS) and direct spectroscopic methods (NMR). A further goal is assessment of the antibacterial properties against methicillin-resistant *Staphylococcus aureus* (MRSA) and *Cutibacterium acnes*, both associated with skin diseases.

## 2. Materials and Methods

### 2.1. Plant Material

　　The aerial parts from *D. communis* were collected from cultivated populations in Stamata of North Attica (Central Greece) (coordinates (WGS84): latitude: 38°08′22.3″ N; longitude: 23°53′09.9″ E) during the flowering stage in June 2019. The collected plant material was recognized and authenticated by Prof. Th. Constantinidis (Faculty of Biology, NKUA). A voucher specimen was deposited in the Department of Pharmacognosy and Chemistry of Natural Products (Faculty of Pharmacy, NKUA) under the code Skaltsa and Rallis 001.

## 2.2. General Experimental Procedures

The NMR spectra were measured in an AVANCE III 600 (Bruker, Corporation, Billerica, MA, USA) instrument equipped with a 5-mm TBI CryoProbe ([1]H-NMR 600 MHz, [13]C-NMR 150 MHz) or a Bruker DRX 400 ([1]H-NMR 400 MHz, Bruker manufacturer, Corporation, Billerica, MA, USA) at 298 K. Chemical shifts were given in ppm ($\delta$) and were referenced to the solvent signals at 3.31/49.0 ppm for MeOD and 7.24/77.0 ppm for CDCl$_3$. Correlation Spectroscopy (COSY), Heteronuclear Single Quantum Correlation (HSQC), and Heteronuclear Multiple Bond Correlation (HMBC) experiments were performed using standard Bruker microprograms. The HRESIMS spectra were obtained with an Agilent MS Q-TOF G6540A spectrometer (Santa Clara, CA, USA). Column chromatography (CC) was performed on a silica gel (Merck, Darmstadt, Germany, Art. 7736; Merck, Art. 9385) or Sephadex LH-20 (Sigma-Aldrich, St. Louis, MO, USA). Fractionation was always monitored by TLC silica gel 60 F-254 (Merck, Darmstadt, Germany, Art. 5554) and cellulose (Merck, Darmstadt, German, Art. 5552) with visualization under UV (254 and 365 nm) and spraying with vanillin-sulfuric acid reagent (Merck, Darmstadt, Germany, Art. S26047 841), as well as Neu's reagent for phenolics (Alfa Aesar, Karlsruhe, Germany, A16606) [22]. Medium-pressure liquid chromatography (MPLC) support consisted of a reversed-phase column (Merck, Darmstadt, Germany, 10167) of 36 × 3.6 cm (Büchi Borosilikat 3.3, Flawil, Switzerland19674) on a system Büchi Pump C-615 and with a flow rate of 1 mL/min. The lyophilizer was Christ, ALPHA I-5 (Apeldoorn, Netherlands). All obtained extracts, fractions, and isolated compounds were evaporated to dryness in a vacuum under low temperature and then were put in activated desiccators with P$_2$O$_5$ until their weights had stabilized.

## 2.3. Extraction, Fractionation, and Isolation

The plant material (berry juice, 1 L) was initially processed by lyophilization to yield a dry residue (50.0 g). Different extraction procedures were applied for further separation or isolation. A part of the lyophilized berry juice (A; 4.0 g) was pre-fractioned by RP$_{18}$-MPLC using mobile phase mixtures of H$_2$O:methanol (MeOH) (from 100:0 to 0:100) of decreasing polarity and gradient elution (flow rate: 1 mL/min) to finally yield 9 fractions of 900 mL each (AA-AI). The fractions were evaporated at reduced pressure below 50 °C, and afterward, they were monitored by [1]H-NMR. Fraction AD (396.4 mg, eluted with H$_2$O:MeOH (80:20)) was subjected to column chromatography (CC) over silica gel using mixtures of cyclohexane (CyHex):dichloromethane and (CH$_2$Cl$_2$):MeOH:H$_2$O of increasing polarity as eluents to give 66 fractions, which were combined based on TLC similarities into 33 groups (AD$_{(A–G)}$). The subfraction AD$_B$ (8.9 mg, eluted with CyHex:CH$_2$Cl$_2$ (30:70)) yielded compound **3** in a mixture with compound **4** and methyl stearate. Fraction AF (60.0 mg, eluted with H$_2$O:MeOH (60:40)) was subjected similarly to CC over silica gel (CyHex:CH$_2$Cl$_2$:MeOH (from 100:0:0:0 to 0:0:100)) to give 33 fractions, which were combined based on TLC similarities into 13 groups (AF$_{(A–M)}$). The subfraction AF$_A$ (10.3 mg, eluted with CyHex:CH$_2$Cl$_2$ (30:70)) yielded compound **3** in a mixture with **4**. Another part of the lyophilized berry juice (A′, 1.65 g) was subjected to further fractionation by CC over silica gel. Mixtures of CyHex, CH$_2$Cl$_2$, MeOH, and H$_2$O with increasing polarity were used as eluents to give 44 fractions, which were combined based on TLC similarities into 20 groups (A′$_{(A–T)}$). Among them, subfraction A′G was identified as compound **4** (5.2 mg). Furthermore, combined subfractions A′L, A′M, and A′N (A′L′; 90.0 mg, eluted with CH$_2$Cl$_2$:MeOH:H$_2$O (60:40:4.0–55:45:4.5)) were subjected to CC over Sephadex LH-20 (MeOH 100%) and yielded 34 fractions, which were combined based on TLC similarities into 12 groups (A′L′$_{(A–L)}$). Subfraction A′L′$_I$ was identified as compound **5** (18.4 mg) [23].

Moreover, two equal amounts (4.0 g) of the lyophilized berry juice (A1 and A2) were subjected to liquid-liquid extraction to extract the non-polar constituents in two different ways. A1 was dissolved in H$_2$O (10 mL), and the aqueous layer was extracted with diethyl ether (Et$_2$O; 10 mL × 3; A1A) and then with CH$_2$Cl$_2$ (10 mL × 3; A1B). A2 was first subjected to acid hydrolysis by boiling under reflux with 10% *w/v* hydrochloric acid (37%

*w/w*) for 120 min, and then it was extracted as described above with Et$_2$O (A2A) and CH$_2$Cl$_2$ (A2B). The organic layers of all extractions were concentrated to dryness, and afterward, they were analyzed by GC-MS and $^1$H-NMR. The $^1$H-NMR analyses revealed the presence of compound **1** in all obtained extracts, while compound **2** was found only in A2A and A2B. In addition, 80.0 mg of the lyophilized berry juice (A3) was extracted with *n*-butanol, and the organic phase, after evaporation, was analyzed by LC-MS. The flow chart of the isolation procedures is shown in Figure S40 [23].

## 2.4. Gas Chromatography-Mass Spectrometry (GC-MS) Analysis

Fractions A1A, A1B, A2A, and A2B, as well as the less polar AE, AG, and AI, were subjected to GC-MS analyses using a Hewlett-Packard 7820A-5977B MSD system (Agilent Technologies, Santa Clara, CA, USA) operating in EI mode (70 eV), equipped with an HP-5MS-fused silica capillary column (30 m × 0.25 mm; film thickness: 0.25 μm) and a split-splitless injector. The temperature program was, from 60 °C at the time of injection, raised to 300 °C at a rate of 3 °C/min and subsequently held at 300 °C for 10 min. Helium was used as a carrier gas at a flow rate of 2.0 mL/min. The injected volume of the samples was 2 μL [24].

The retention indices for all compounds were determined according to the Van der Dool approach [25], with reference to a homologous series of *n*-alkanes from C$_9$ to C$_{25}$. The identification of the chemical components was based on a comparison of both relative retention times and mass spectra with those reported by Adams [26] and the NIST/NBS and Wiley libraries. The component relative percentages were calculated based on the GC peak areas without using correction factors [24].

## 2.5. Liquid Chromatography High-Resolution Quadrupole Time-of-Flight Mass Spectrometry (LC-Q-TOF-MS/MS)

Analyses of the butanol extract of berry juice (A3) and of selected fractions were performed with a UHPLC Agilent 1290 infinity system with a DAD G4212A and MS Agilent G6540A Q-TOF with Agilent Jet Stream technology electrospray ionization. Separation was performed on a Phenomenex Luna Omega column (C18, 1.9 u, 90 A°, 75 × 2.0 mm) using gradient mixtures of 0.1% formic acid (solvent A) and MeCN supplemented with 0.1% formic acid (solvent B) (gradient: 0.0−8.0 min, 0%→30% B; 8.0−8.1 min, 30%→98% B; 8.1−9.1 min, 98% B; 9.1−9.2 min, 98%→5% B; 9.2−10.0 min, 5% B; flow rate: 0.6 mL/min; injection volume: 1 μL; oven temperature: 40 °C). Data analysis was performed by MassHunter Workstation Software Qualitative Analysis (B.07.00, Agilent) using automatic mass spectrum integration. LC-Q-TOF-MS/MS analyses were performed in positive and negative ionization modes to obtain the maximum information on its composition. The metabolites were characterized based on their mass spectra using the precursor ion and comparison of the fragmentation patterns with molecules described in the literature [27].

## 2.6. Nuclear Magnetic Resonance Spectroscopy (NMR) Spectroscopy

During the whole analysis course, all extracts and obtained subfractions were continuously monitored and traced down using an NMR metabolomic strategy, which permitted detailed characterization thereof. Furthermore, the NMR spectra of compounds **1–5** were measured (Figures S19–S33), as well as of the fractions with low complexity (AE, AG, and AI).

## 2.7. Identification of Cutibacterium acnes Strain ATCC 6919 by 16S rRNA Gene Sequencing

Genomic DNA extraction from Pure BHI broth and Anaerobe CDC Blood agar cultures of *Cutibacterium acnes* strain ATCC 6919 was performed using an ExtractMe Genomic DNA Kit (Blirt, Gdánsk, Poland). Universal primers 27F (5′-AGAGTTTGATCMTGGCTCAG-3′) [28] and 1492R (5′-GGTTACCTTGTTACGACTT-3′) [29] (Eurofins Genomics, Germany) were used to amplify the 16S rRNA gene by PCR. The reaction mixture contained the following: 1 U FastGene Taq DNA Polymerase (NIPPON Genetics, Tokyo, Japan), 1 × PCR buffer A,

25 pmol of each primer, 1 mM dNTPs, a 3μL DNA template, and deionized sterile water at a final volume of 50 μL. The thermal cycler Primus 25 (PEQLAB Biotechnologie, Erlangen, Germany) was used in the following PCR conditions: initialization at 95 °C for 3 min, followed by 35 cycles of denaturation at 95 °C for 30 s, annealing at 50 °C for 30 s, and elongation at 72 °C for 2 min. A final elongation step at 72 °C for 5 min was added.

Amplicons of *C. acnes* were purified using the NucleoSpin Gel and PCR clean-up kit (Macherey-Nagel, Düren, Germany) and then directly sequenced via the Sanger dideoxy termination method by Cemia (Larissa, Greece). Chromas (Version 2.6.6 Software, Technelysium Pty Ltd., South Brisbane, Australia, www.technelysium.com.au, accessed on 20 October 2021) was used to check the quality of the obtained sequencing results. The sequences were assembled into a single sequence via MEGA X (Version 10.1.6 Software) [30] and Gene Runner (Version 6.5 Software, Inc., Hudson, NY, USA, www.generunner.net accessed on 20 October 2021) and subjected to a BlastN (Megablast) (Bethesda, MD, USA, https://blast.ncbi.nlm.nih.gov/Blast.cgi, accessed on 20 October 2021 ) search in the 16S rRNA Database-GENEBANK to identify the sequences with the highest similarity.

### 2.8. Bacterial Strains and Growth Conditions

The antibacterial activity of *D. communis* berry juice was determined against MRSA strain 1552 and *Cutibacterium acnes* strain ATCC 6919. MRSA strain 1552 was isolated from the clinical samples, and the identification and characterization were conducted by standard laboratory methods (kindly provided by Prof. Spyros Pournaras, School of Medicine, NKUA). MRSA was routinely grown in Müller–Hinton broth (Lab M, Bury, UK) or Müller–Hinton agar (Lab M, Bury, UK) at 37 °C aerobically and *C. acnes* in Brain Heart Infusion (BHI) broth (Condalab S.A., Spain) or BHI agar (Condalab S.A., Spain) at 37 °C anaerobically.

### 2.9. Determination of Minimum Inhibitory Concentration (MIC)

The minimum inhibitory concentration (MIC) of the berry juice and AC-AG fractions were determined in sterile 96-well polystyrene microtiter plates (Kisker Biotech GmbH & Co. KG, Steinfurt, Germany) using a spectrophotometric bioassay as previously described [31], with some modifications. Briefly, 0.25 g of berry juice was suspended in sterile ddH$_2$O (2-mL final volume) for 1 h at room temperature with occasional vortexing and then centrifuged at $10,000 \times g$ for 7 min. The aqueous phase was filtered through a 0.22-μm syringe filter and used for serial dilutions (in Müller–Hinton and BHI broth for MRSA and *C. acnes*, respectively), corresponding from 25 to 0.39% *w/v*. The weighed part of the AC-AG fractions was suspended in sterile ddH$_2$O containing 1.5% DMSO (2.5-mL final volume) and then centrifuged at $5000 \times g$ for 3 min. The aqueous phase was filtered through a 0.22-μm syringe filter and used for serial dilutions as described above. Overnight bacterial cultures of MRSA (grown in Müller–Hinton) were adjusted to a 0.5 McFarland turbidity standard (~$1.5 \times 10^8$ CFU/mL). For 3 days, the old bacterial cultures of *C. acnes* (grown in BHI broth) were adjusted to a 0.5 McFarland turbidity standard (~$1.5 \times 10^8$ CFU/mL). A 10-μL broth, containing approximately $5 \times 10^4$ CFUs, was added to 190 μL of the tested twofold sample dilutions.

The positive control wells, containing broth, were inoculated with MRSA or *C. acnes* to test the growth of the pathogen. The negative control wells contained dilutions of berry juice or fractions in Müller–Hinton or BHI broth without bacteria. The Müller–Hinton or BHI broth control wells without bacteria were used to test for any possible contamination.

The optical density (OD) was determined at 600 nm using an EL × 808 absorbance microplate reader (BioTek Instruments, Inc., Winooski, VT, USA) just prior to incubation (t = 0) and 24 h after incubation (t = 24 h) at 37 °C aerobically for MRSA (t = 0) and 5 days after incubation (t = 5 d) at 37 °C for *C. acnes* under anaerobic conditions. The OD for each negative control replicate well (containing sample) at t = 24 or t = 5 d for MRSA and *C. acnes* was subtracted from the OD of the same replicate test well with bacteria at t = 24 or t = 5 d for MRSA and *C. acnes*, respectively. The growth inhibition at each sample dilution was determined using the formula

% inhibition = [1 − (OD test well − OD of corresponding negative control well)] × 100. The MIC was determined as the lowest sample concentration which resulted in 100% growth inhibition. The MIC values of the berry juice and AC-AG fractions were expressed as $w/v$ and mg/mL, respectively.

*2.10. Determination of Minimum Bactericidal Concentration (MBC)*

The MBC was determined by transferring a small quantity of the sample contained in each replicate well of the microtiter plates to Müller–Hinton agar plates for MRSA and BHI agar for *C. acnes* by using a microplate replicator (Boekel Scientific, Feasterville-Trevose, PA, USA). The plates were incubated for 24 h aerobically for MRSA and 5 days anaerobically for *C. acnes* at 37 °C. The MBC was determined as the lowest concentration of the initial extract and fractions at which no grown colonies were observed [32].

**3. Results and Discussion**

Previous studies on the title plant showed that root tubers have been the most intensively investigated plant part, with triterpenoids, sterols, and saponins as well as phenanthrenes and furanocoumarins being reported [11,12,33,34]. Similarly, the aerial parts (leaves and shoots) have resulted in a different yield of phenolic derivatives and flavonoids, saponins, sterols, triterpenoids, carotenoids, tocopherols, fatty acids, and organic acids [11,15,35,36]. Nevertheless, to the best of our knowledge, extracts from the berry juice led to the identification of sterols and flavonoids in the *O*- and *C*-glucoside forms, though no extensive phytochemical investigation has been performed [16,37]. In southern Central Greece, *D. communis* berry juice has been reported (by oral sources) as a traditional remedy for skin-related ailments, as well as for oral infections. Taking into consideration the above, our study was oriented to the investigation of the berry juice, targeting its phytochemical content and its potential biological effects on bacteria related to skin infections.

The phytochemical analysis of *D. communis* berry juice was processed by different chromatographic techniques. Through GC-MS and LC-MS/MS chromatographies aided by NMR spectroscopy, a great number of compounds belonging to different phytochemical groups was identified. The results are presented in Tables 1–8.

**Table 1.** Chemical composition of diethyl ether extract of *Dioscorea communis* berry juice (A1A).

| No. | Retention Time | % Area | KI | AI | Name of Compound | Molecular Formula | MW |
|-----|---------------|--------|------|------|------------------|-------------------|-----|
| 1 | 39.611 | 6.3 | 1950 | 1964 | 1,2-benzenedicarboxylic acid, dibutyl ester [dibutyl phthalate] | $C_{16}H_{22}O_4$ | 278 |
| 2 | 41.018 | 3.0 | 1993 | 1992 | hexadecanoic acid, ethyl ester [ethyl palmitate] | $C_{18}H_{36}O_2$ | 284 |
| 3 | 44.032 | 2.1 | 2089 | 2095 | 9Z,12Z-octadecadienoic acid, methyl ester [methyl linoleate] | $C_{19}H_{34}O_2$ | 294 |
| 4 | 46.097 | 45.2 | 2157 | 2159 | 9Z,12Z-octadecadienoic acid, ethyl ester [ethyl linoleate] | $C_{20}H_{36}O_2$ | 308 |
| 5 | 46.272 | 36.9 | 2163 | 2173 | 9Z,12Z,5Z-octadecatrienoic acid, ethyl ester [ethyl linolenate] | $C_{20}H_{34}O_2$ | 306 |
| 6 | 46.515 | 3.7 | 2171 | 2179 | 9Z-octadecenoic acid, ethyl ester [ethyl oleate] | $C_{20}H_{38}O_2$ | 310 |
| 7 | 70.332 | 2.8 | | 3130 | *α*-tocopherol | $C_{29}H_{50}O_2$ | 430 |
| | Total | 100.0 | | | | | |

**Table 2.** Chemical composition of dichloromethane extract of *Dioscorea communis* berry juice (A1B).

| No. | Retention Time | % Area | KI | AI | Name of Compound | Molecular Formula | MW |
|-----|---------------|--------|------|------|-----------------|-------------------|-----|
| 1 | 41.014 | 3.3 | 1993 | 1992 | hexadecanoic acid, ethyl ester (ethyl palmitate) | $C_{18}H_{36}O_2$ | 284 |
| 2 | 44.025 | 2.7 | 2089 | 2095 | 9Z,12Z-octadecadienoic acid, methyl ester (methyl linoleate) | $C_{19}H_{34}O_2$ | 294 |
| 3 | 44.206 | tr | 2094 | 2105 | 9Z,12Z,15Z-octadecatrienoic acid, methyl ester (methyl linolenate) | $C_{19}H_{32}O_2$ | 292 |
| 4 | 46.095 | 46.7 | 2157 | 2159 | 9Z,12Z-octadecadienoic acid, ethyl ester (ethyl linoleate) | $C_{20}H_{36}O_2$ | 308 |
| 5 | 46.271 | 38.2 | 2163 | 2173 | 9Z,12Z,15Z-octadecatrienoic acid, ethyl ester (ethyl linolenate) | $C_{20}H_{34}O_2$ | 306 |
| 6 | 46.509 | 3.6 | 2170 | 2179 | 9Z-octadecenoic acid, ethyl ester (ethyl oleate) | $C_{20}H_{38}O_2$ | 310 |
| 7 | 70.328 | 2.9 | | 3130 | α-tocopherol | $C_{29}H_{50}O_2$ | 430 |
| 8 | 73.784 | 2.6 | | 3203 | (3β)-stigmast-5-en-3-ol (β-sitosterol) | $C_{29}H_{50}O$ | 414 |
| | Total | 100.0 | | | | | |

**Table 3.** Chemical composition of diethyl ether extract of *Dioscorea communis* berry juice after acid hydrolysis (A2A).

| No. | Retention Time | % Area | KI | AI | Name of Compound | Molecular Formula | MW |
|-----|---------------|--------|------|------|-----------------|-------------------|-----|
| 1 | 37.969 | tr | 1900 | 1890 | 9Z-hexadecenoic acid, methyl ester (methyl palmitoleate) | $C_{17}H_{32}O_2$ | 268 |
| 2 | 38.823 | 53.3 | 1926 | 1921 | hexadecanoic acid, methyl ester (methyl palmitate) | $C_{17}H_{34}O_2$ | 270 |
| 3 | 44.038 | 20.3 | 2089 | 2095 | 9Z,12Z-octadecadienoic acid, methyl ester (methyl linoleate) | $C_{19}H_{34}O_2$ | 294 |
| 4 | 44.202 | 14.8 | 2094 | 2105 | 9Z,12Z,15Z-octadecatrienoic acid, methyl ester (methyl linolenate) | $C_{19}H_{32}O_2$ | 292 |
| 5 | 44.435 | 1.3 | 2102 | 2103 | 9Z-octadecenoic acid, methyl ester (methyl oleate) | $C_{19}H_{36}O_2$ | 296 |
| 6 | 45.142 | 3.5 | 2125 | 2124 | octadecanoic acid, methyl ester (methyl stearate) | $C_{19}H_{38}O_2$ | 298 |
| 7 | 46.087 | 3.2 | 2156 | 2159 | 9Z,12Z-octadecadienoic acid, ethyl ester (ethyl linoleate) | $C_{20}H_{36}O_2$ | 308 |
| 8 | 46.274 | 2.2 | 2163 | 2173 | 9Z,12Z,15Z-octadecatrienoic acid, ethyl ester (ethyl linolenate) | $C_{20}H_{34}O_2$ | 306 |
| 9 | 47.570 | 1.5 | 2206 | | unknown ($m/z$ = 278.3) | | |
| 10 | 51.039 | tr | 2324 | 2329 | eicosanoic acid, methyl ester (methyl arachidate) | $C_{21}H_{42}O_2$ | 326 |
| | Total | 98.5 | | | | | |

**Table 4.** Chemical composition of dichloromethane extract of *Dioscorea communis* berry juice after acid hydrolysis (A2B).

| No. | Retention Time | % Area | KI | AI | Name of Compound | Molecular Formula | MW |
|-----|---------------|--------|------|------|-----------------|-------------------|-----|
| 1 | 38.823 | 33.2 | 1926 | 1921 | hexadecanoic acid, methyl ester (methyl palmitate) | $C_{17}H_{34}O_2$ | 270 |
| 2 | 41.061 | tr | 1995 | 1992 | hexadecanoic acid, ethyl ester (ethyl palmitate) | $C_{18}H_{36}O_2$ | 284 |
| 3 | 44.059 | 7.7 | 2090 | 2095 | 9Z,12Z-octadecadienoic acid, methyl ester (methyl linoleate) | $C_{19}H_{34}O_2$ | 294 |
| 4 | 44.280 | 6.8 | 2097 | 2105 | 9Z,12Z,15Z-octadecatrienoic acid, methyl ester (methyl linolenate) | $C_{19}H_{32}O_2$ | 292 |
| 5 | 45.217 | tr | 2128 | 2124 | octadecanoic acid, methyl ester (methyl stearate) | $C_{19}H_{38}O_2$ | 298 |
| 6 | 46.117 | 28.1 | 2157 | 2159 | 9Z,12Z-octadecadienoic acid, ethyl ester (ethyl linoleate) | $C_{20}H_{36}O_2$ | 308 |
| 7 | 46.315 | 24.2 | 2164 | 2173 | 9Z,12Z,15Z-octadecatrienoic acid, ethyl ester (ethyl linolenate) | $C_{20}H_{34}O_2$ | 306 |
| | Total | 100.0 | | | | | |

**Table 5.** Chemical composition of fraction AI.

| No. | Retention Time | % Area | KI | AI | Name of Compound | Molecular Formula | MW |
|---|---|---|---|---|---|---|---|
| 1 | 7.787 | 2.3 | 1102 | 1100 | nonanal (pelargonaldehyde) | $C_9H_{18}O$ | 142 |
| 2 | 27.237 | 5.3 | 1598 | 1600 | hexadecane | $C_{16}H_{34}$ | 226 |
| 3 | 34.204 | 1.9 | 1791 | 1803 | 3-hexadecanone | $C_{16}H_{32}O$ | 240 |
| 4 | 34.930 | 4.9 | 1812 | 1811 | hexadecanal (palmitaldehyde) | $C_{16}H_{32}O$ | 240 |
| 5 | 35.956 | 1.6 | 1842 | 1845 | 6,10,14-trimethyl-2-pentadecanone (hexahydrofarnesyl acetone) | $C_{18}H_{36}O$ | 268 |
| 6 | 38.238 | 9.4 | 1909 | 1902 | 2*E*-nonadecene | $C_{19}H_{38}$ | 266 |
| 7 | 38.398 | 5.0 | 1914 | | unknown (*m/z* = 266.1) | | |
| 8 | 38.750 | 2.4 | 1924 | 1921 | hexadecanoic acid, methyl ester (methyl palmitate) | $C_{17}H_{34}O_2$ | 270 |
| 9 | 41.213 | 34.0 | 1999 | 2004 | 2-octadecanone | $C_{18}H_{36}O$ | 268 |
| 10 | 41.724 | 3.0 | 2016 | 2013 | octadecanal (stearaldehyde) | $C_{18}H_{36}O$ | 268 |
| 11 | 43.737 | 1.3 | 2079 | 2077 | 1-octadecanol (stearyl alcohol) | $C_{18}H_{38}O$ | 270 |
| 12 | 44.068 | 1.7 | 2090 | 2095 | 9Z,12Z-octadecadienoic acid, methyl ester (methyl linoleate) | $C_{19}H_{34}O_2$ | 294 |
| 13 | 44.242 | 2.6 | 2095 | 2103 | 9Z-octadecenoic acid, methyl ester (methyl oleate) | $C_{19}H_{36}O_2$ | 296 |
| 14 | 45.053 | 3.0 | 2122 | | unknown (*m/z* = 282.1) | | |
| 15 | 45.133 | 2.2 | 2125 | 2124 | octadecanoic acid, methyl ester (methyl stearate) | $C_{19}H_{38}O_2$ | 298 |
| 16 | 45.667 | 1.5 | 2143 | 2141 | 9Z-octadecenoic acid (oleic acid) | $C_{18}H_{34}O_2$ | 282 |
| 17 | 46.618 | 2.9 | 2174 | | unknown (*m/z* = 283.3) | | |
| 18 | 47.256 | 2.3 | 2195 | 2196 | octadecanoic acid, ethyl ester (ethyl stearate) | $C_{20}H_{40}O_2$ | 312 |
| 19 | 47.395 | 6.3 | 2200 | 2200 | docosane | $C_{22}H_{46}$ | 310 |
| 20 | 50.300 | 1.2 | 2298 | 2300 | tricosane | $C_{23}H_{48}$ | 324 |
| 21 | 51.307 | 0.9 | 2332 | 2329 | eicosanoic acid, methyl ester (methyl arachidate) | $C_{21}H_{42}O_2$ | 326 |
| 22 | 51.720 | 2.0 | 2347 | | unknown (*m/z* = 323.3) | | |
| 23 | 53.078 | 2.3 | 2393 | 2400 | tetracosane | $C_{24}H_{50}$ | 338 |
| | Total | 89.1 | | | | | |

**Table 6.** Chemical composition of fraction AE.

| No. | Retention Time | % Area | KI | AI | Name of Compound | Molecular Formula | MW |
|---|---|---|---|---|---|---|---|
| 1 | 34.435 | 7.1 | 1797 | 1800 | 2-hexadecanone | $C_{16}H_{32}O$ | 240 |
| 2 | 35.344 | 1.7 | 1824 | 1826 | pentadecanoic acid, methyl ester | $C_{16}H_{32}O_2$ | 256 |
| 3 | 37.890 | 3.2 | 1898 | 1901 | 2-heptadecanone | $C_{17}H_{34}O$ | 254 |
| 4 | 38.767 | 3.8 | 1925 | 1921 | hexadecanoic acid, methyl ester (methyl palmitate) | $C_{17}H_{34}O_2$ | 270 |
| 5 | 41.437 | 49.4 | 2007 | 2004 | 2-octadecanone | $C_{18}H_{36}O$ | 268 |
| 6 | 42.043 | 1.7 | 2026 | 2028 | heptadecanoic acid, methyl ester (methyl margarate) | $C_{18}H_{38}O_2$ | 284 |
| 7 | 44.112 | 4.2 | 2091 | 2095 | 9Z,12Z-octadecadienoic acid, methyl ester (methyl linoleate) | $C_{19}H_{34}O_2$ | 294 |
| 8 | 45.180 | 20.8 | 2126 | 2124 | octadecanoic acid, methyl ester (methyl stearate) | $C_{19}H_{38}O_2$ | 298 |
| 9 | 45.713 | 4.8 | 2144 | 2141 | 9Z-octadecenoic acid (oleic acid) | $C_{18}H_{34}O_2$ | 282 |
| 10 | 47.281 | 3.3 | 2196 | | unknown (*m/z* = 313.2) | | |
| | Total | 96.7 | | | | | |

**Table 7.** Chemical composition of fraction AG.

| No. | Retention Time | % Area | KI | AI | Name of Compound | Molecular Formula | MW |
|---|---|---|---|---|---|---|---|
| 1 | 34.928 | tr | 1812 | 1811 | hexadecanal (palmitaldehyde) | $C_{16}H_{32}O$ | 240 |
| 2 | 37.855 | 0.6 | 1897 | 1901 | 2-heptadecanone | $C_{17}H_{34}O$ | 254 |
| 3 | 38.228 | 2.5 | 1909 | 1902 | 2*E*-nonadecene | $C_{19}H_{38}$ | 266 |
| 4 | 38.358 | 1.6 | 1913 | | unknown (*m/z* = 266.1) | | |
| 5 | 38.734 | 1.3 | 1924 | 1921 | hexadecanoic acid, methyl ester (methyl palmitate) | $C_{17}H_{34}O_2$ | 270 |
| 6 | 40.385 | 1.0 | 1975 | | unknown (*m/z* = 285.1) | | |

**Table 7.** *Cont.*

| No. | Retention Time | % Area | KI | AI | Name of Compound | Molecular Formula | MW |
|-----|----------------|--------|------|------|------------------|-------------------|-----|
| 7 | 41.007 | tr | 1994 | 1992 | hexadecanoic acid, ethyl ester (ethyl palmitate) | $C_{18}H_{36}O_2$ | 284 |
| 8 | 41.372 | 59.7 | 2005 | 2004 | 2-octadecanone | $C_{18}H_{36}O$ | 268 |
| 9 | 41.690 | 2.4 | 2015 | 2013 | octadecanal (stearaldehyde) | $C_{18}H_{36}O$ | 268 |
| 10 | 42.013 | 0.7 | 2025 | 2028 | heptadecanoic acid, methyl ester (methyl margarate) | $C_{18}H_{38}O_2$ | 284 |
| 11 | 42.778 | 1.4 | 2050 | | unknown ($m/z = 282.1$) | | |
| 12 | 43.568 | 1.6 | 2075 | | unknown ($m/z = 299.1$) | | |
| 13 | 45.163 | 11.5 | 2126 | 2124 | octadecanoic acid, methyl ester (methyl stearate) | $C_{19}H_{38}O_2$ | 298 |
| 14 | 46.080 | 0.4 | 2157 | 2159 | 9Z,12Z-octadecadienoic acid, ethyl ester (ethyl linoleate) | $C_{20}H_{36}O_2$ | 308 |
| 15 | 46.646 | 11.6 | 2175 | | unknown ($m/z = 313.3$) | | |
| 16 | 47.386 | 0.5 | 2200 | 2200 | docosane | $C_{22}H_{46}$ | 310 |
| 17 | 71.869 | 0.5 | | 3131 | (3$\beta$,24R)-ergost-5-en-3-ol (campesterol) | $C_{28}H_{48}O$ | 410 |
| 18 | 72.575 | 0.8 | | 3170 | (3$\beta$,22E)-stigmasta-5,22-dien-3-ol (stigmasterol) | $C_{29}H_{48}O$ | 412 |
| 19 | 73.781 | 1.9 | | 3203 | (3$\beta$)-stigmast-5-en-3-ol ($\beta$-sitosterol) | $C_{29}H_{50}O$ | 414 |
| | Total | 82.8 | | | | | |

Components are listed in all tables according to their elution from an HP-5MS column. KI = Kováts indices calculated against $C_9$-$C_{25}$ *n*-alkanes on the HP-5MS column; AI = arithmetic indices; and tr = traces.

**Table 8.** LC-MS of butanol extract.

| Positive Ion Mode | | | Negative Ion Mode | | Molecular Formula | Metabolite Name |
|-------------------|-------|-------|-------------------|-------|-------------------|-----------------|
| RT | Found | Calcd | Found | Calcd | | |
| 0.324 | 203.0530 [M + Na]$^+$ | 203.0526 | 179.0563 [M − H]$^-$ | 179.0561 | $C_6H_{12}O_6$ | hexose |
| 0.337 | 365.1059 [M + Na]$^+$ | 365.1054 | 341.1093 [M − H]$^-$ | 341.1089 | $C_{12}H_{22}O_{11}$ | hexose-pentose |
| 0.488 | 113.0217 [M + Na]$^+$ | 113.0209 | 89.0243 [M − H]$^-$ | 89.0244 | $C_3H_6O_3$ | lactic acid |
| 0.490 | 349.1121 [M + Na]$^+$ | 349.1105 | 371.1189 [M + HCOO]$^-$ | 371.1195 | $C_{12}H_{22}O_{10}$ | hexose-pentose |
| 0.559 | | | 175.025 [M − H]$^-$ | 175.0248 | $C_6H_8O_6$ | ascorbic acid |
| 0.565 | 117.0182 [M + H]$^+$ | 117.0182 | | | $C_4H_4O_4$ | fumaric acid |
| 0.605 | 121.0652 [M + H]$^+$ | 121.0648 | | | $C_8H_8O$ | 2-methylbenzaldehyde |
| 0.917 | 166.0866 [M + H]$^+$ | 166.0863 | 164.0716 [M − H]$^-$ | 164.0717 | $C_9H_{11}NO_2$ | phenylalanine |
| 1.208 | 146.0604 [M + H]$^+$ | 146.0600 | | | $C_9H_7NO$ | 4-formyl indole |
| 1.210 | 208.0609 [M + H]$^+$ | 208.0604 | 206.0458 [M − H]$^-$ | 206.0459 | $C_{10}H_9NO_4$ | pyranonigrin S |
| 1.303 | 239.1489 [M + H]$^+$ | 239.1489 | | | $C_{10}H_{22}O_6$ | unknown |
| 1.404 | 188.0707 [M + H]$^+$ | 188.0706 | | | $C_{11}H_9NO_2$ | unknown |
| 1.725 | 247.1080 [M + H]$^+$ | 247.1077 | | | $C_{13}H_{14}N_2O_3$ | unknown |
| 1.794 | 217.0977 [M + H]$^+$ | 217.0972 | | | $C_{12}H_{12}N_2O_2$ | unknown |
| 1.973 | 231.1131 [M + H]$^+$ | 231.1128 | 229.0983 [M − H]$^-$ | 229.0983 | $C_{13}H_{14}N_2O_2$ | unknown |
| 1.983 | 611.1609 [M + H]$^+$ | 611.1607 | 609.1462 [M − H]$^-$ | 609.1461 | $C_{27}H_{30}O_{16}$ | rutin |
| 2.046 | 181.0498 [M + H]$^+$ | 181.0495 | 179.0350 [M − H]$^-$ | 179.0350 | $C_9H_8O_4$ | caffeic acid |
| 2.175 | 275.1032 [M + H]$^+$ | 275.1026 | | | $C_{14}H_{14}N_2O_4$ | unknown |
| 2.183 | 595.1668 [M + H]$^+$ | 595.1657 | 593.1516 [M − H]$^-$ | 593.1512 | $C_{27}H_{30}O_{15}$ | kaempferol 3-*O*-rutinoside |
| 2.490 | | | 165.0556 [M − H]$^-$ | 165.0557 | $C_9H_{10}O_3$ | phenolic |
| 3.006 | 349.1646 [M + H]$^+$ | 349.1646 | 347.1498 [M − H]$^-$ | 347.1500 | $C_{19}H_{24}O_6$ | unknown |
| 3.248 | | | 345.1341 [M − H]$^-$ | 345.1644 | $C_{19}H_{22}O_6$ | unknown |
| 4.183 | 883.4695 [M + H]$^+$ | 883.4686 | | | $C_{45}H_{70}O_{17}$ | 7-oxodioscin |
| 4.192 | 399.1781 [M + Na]$^+$ | 399.1778 | 375.181 [M − H]$^-$ | 375.1813 | $C_{21}H_{28}O_6$ | 3,5-dihydroxy-1,7-bis(4-hydroxy-3-methoxyphenyl)heptane |
| 4.204 | 303.0502 [M + H]$^+$ | 303.0499 | 301.0353 [M − H]$^-$ | 301.0354 | $C_{15}H_{10}O_7$ | quercetin |
| 4.714 | 279.2325 [M + H]$^+$ | 279.2319 | | | $C_{18}H_{30}O_2$ | linolenic acid |
| 4.874 | 287.0551 [M + H]$^+$ | 287.0550 | 285.0406 [M − H]$^-$ | 285.0405 | $C_{15}H_{10}O_6$ | kaempferol |
| 5.050 | | | 329.2333 [M − H]$^-$ | 329.2333 | $C_{18}H_{34}O_5$ | 9,10,11-trihydroxy-12-octadecenoic acid |
| 6.060 | 312.2533 [M + H]$^+$ | 312.2533 | | | $C_{18}H_{33}NO_3$ | unknown |
| 6.458 | 885.4842 [M + H]$^+$ | 885.4842 | 929.4747 [M + HCOO]$^-$ | 929.4752 | $C_{45}H_{72}O_{17}$ | gracillin |
| 6.524 | 315.2537 [M + H]$^+$ | 315.2530 | 313.2383 [M − H]$^-$ | 313.2384 | $C_{18}H_{34}O_4$ | lipid derivative |

**Table 8.** *Cont.*

| Positive Ion Mode | | | Negative Ion Mode | | Molecular Formula | Metabolite Name |
|---|---|---|---|---|---|---|
| RT | Found | Calcd | Found | Calcd | | |
| 6.805 | 415.2117 [M + H]$^+$ | 415.2115 | | | $C_{24}H_{30}O_6$ | bersenogenin |
| 6.899 | 478.2943 [M + H]$^+$ | 478.2952 | 476.2794 [M − H]$^-$ | 476.2806 | $C_{30}H_{39}NO_4$ | 18-deoxycytochalasin H |
| 7.044 | 339.2509 [M + Na]$^+$ | 339.2506 | 315.2541 [M − H]$^-$ | 315.2541 | $C_{18}H_{36}O_4$ | lipid derivative |
| 7.086 | 310.2381 [M + H]$^+$ | 310.2377 | 308.2232 [M − H]$^-$ | 308.2231 | $C_{18}H_{31}NO_3$ | lipid amide |
| 7.296 | 492.1134 [M + NH$_4$]$^+$ | 492.1137 | 473.0724 [M − H]$^-$ | 473.0725 | $C_{22}H_{18}O_{12}$ | cichoric acid |
| 7.539 | 891.4708 [M + Na]$^+$ | 891.4713 | 913.4790 [M+HCOO]$^-$ | 913.4802 | $C_{45}H_{72}O_{16}$ | dioscin |
| 7.657 | 280.2640 [M + H]$^+$ | 280.2635 | | | $C_{18}H_{33}NO$ | linoleamide |
| 7.672 | 449.3732 [M + Na]$^+$ | 449.3757 | | | $C_{30}H_{50}O$ | amyrin |
| 7.782 | | | 295.2280 [M − H]$^-$ | 295.2279 | $C_{18}H_{32}O_3$ | 9-hydroxy-10,12-octadecadienoic acid |
| 7.887 | 257.2478 [M + H]$^+$ | 257.2475 | 255.2329 [M − H]$^-$ | 255.2330 | $C_{16}H_{32}O_2$ | palmitic acid |
| 7.970 | | | 358.2595 [M − H]$^-$ | 358.2599 | $C_{19}H_{37}NO_5$ | unknown |
| 7.990 | 296.2592 [M + H]$^+$ | 296.2584 | | | $C_{18}H_{33}NO_2$ | stearimide |
| 8.076 | 437.3729 [M + Na]$^+$ | 437.3754 | | | $C_{29}H_{50}O$ | β-sitosterol |
| 8.235 | 685.4366 [M + Na]$^+$ | 685.4356 | | | $C_{42}H_{63}O_4P$ | tris-(2,4-di-*tert*-butylphenyl)phosphate |
| 8.251 | 239.2377 [M + H]$^+$ | 239.2397 | | | $C_{16}H_{30}O$ | 2-hexadecenal |
| 8.248 | 281.2481 [M + H]$^+$ | 281.2475 | 279.2327 [M − H]$^-$ | 279.2330 | $C_{18}H_{32}O_2$ | linoleic acid |
| 8.400 | 453.3685 [M + Na]$^+$ | 453.3703 | | | $C_{29}H_{50}O_2$ | α-tocopherol |
| 8.704 | 228.2329 [M + H]$^+$ | 228.2322 | | | $C_{14}H_{29}NO$ | myristamide |
| 9.025 | 254.2484 [M + H]$^+$ | 254.2478 | | | $C_{16}H_{31}NO$ | palmitoleamide |
| 9.550 | | | 271.2277 [M − H]$^-$ | 271.2279 | $C_{16}H_{32}O_3$ | unknown |
| 9.689 | 307.2628 [M + H]$^+$ | 307.2632 | 351.2539 [M+HCOO]$^-$ | 351.2541 | $C_{20}H_{34}O_2$ | ethyl linolenate |
| 9.868 | 256.2638 [M + H]$^+$ | 256.2625 | | | $C_{16}H_{33}NO$ | palmitamide |
| 10.109 | 282.2791 [M + H]$^+$ | 282.2791 | | | $C_{18}H_{35}NO$ | 9-octadecenamide |
| 10.147 | 309.2787 [M + H]$^+$ | 309.2788 | 353.2694 [M + HCOO]$^-$ | 353.2697 | $C_{20}H_{36}O_2$ | ethyl linoleate |
| 11.58 | 285.2794 [M + H]$^+$ | 285.2788 | 283.2643 [M − H]$^-$ | 283.2643 | $C_{18}H_{36}O_2$ | ethyl palmitate |

### 3.1. Chemical Composition by GC-MS Analyses in Various Fractions of Dioscorea communis Berry Juice

At first, the chemical composition of the berry juice was achieved by two approaches. Extracts with different polarities were obtained through liquid-liquid extraction (see Section 2.3 and Figures S1–S18). The untreated (A1A and A1B) and after acid hydrolysis non-polar extracts (A2A and A2B) were submitted to GC-MS analyses, revealing the presence of several fatty acid esters (Tables 1–4). It is noteworthy that both qualitative and quantitative differences were observed after acid hydrolysis, which could be attributed to the hydrolysis of fatty acid esters, triglycerides, or phospholipids [38]. The chemical fingerprints of the untreated Et$_2$O (A1A) and CH$_2$Cl$_2$ (A1B) extracts were quite similar, with ethyl linoleate (45.2% and 46.7%, respectively) and ethyl linolenate (36.9% and 38.1%, respectively) being their main metabolites. After acid hydrolysis, the obtained Et$_2$O extract (A2A) was characterized by the presence of methyl esters of palmitic (53.3%), linoleic (20.3%), and linolenic (14.8%) acid, while the CH$_2$Cl$_2$ extract (A2B) was abundant in methyl palmitate (33.2%), ethyl linoleate (28.1%), and ethyl linolenate (24.2%). It was noticed that the ethyl esters of linoleic and linolenic acid were present in all extracts, while methyl linolenate was absent in the untreated Et$_2$O extract (A1A). Phthalates, such as dibutyl phthalate (Table 1, compound **1**), are used as plasticizer solvents. Thus far, they have been previously described from the genus *Dioscorea* and the family Dioscoreaceae, as well as from other natural sources [39]. However, their presence as natural products is controversial, as they could be either stored from the environment or co-extracted using solvents during the handling of the plant material [40].

For a more detailed analysis, part of the lyophilized berry juice was subjected to RP$_{18}$-MPLC, and the yielded fractions were screened by $^1$H-NMR. Based on the obtained spectra, three fractions (AI, AE, and AG) were selected and further analyzed by GC-MS (Tables 5–7). Briefly, the fractions AI, AE, and AG were mixtures of fatty acid esters, ketones, aldehydes, alcohols, and hydrocarbons. In detail, 23 compounds were detected in fraction AI, with 2-octadecanone (34.0%) and 2*E*-nonadecene (9.4%) being the main constituents.

In fraction AE, 10 compounds were detected, and once more the main ingredient proved to be 2-octadecanone (49.4%), while methyl stearate was also abundant (20.8%). Fraction AG featured 19 compounds, and again the predominant compound was 2-octadecanone (59.7%), followed by methyl stearate (11.5%) and an unknown compound (11.6%), with $m/z = 313.3$. The available GC-MS libraries do not include data regarding *N*- and *P*-containing compounds. However, the odd $m/z$ values suggested the presence of such compounds, which was further supported by the LC-MS analysis.

### 3.2. NMR Analyses in Non-Polar Fractions of Dioscorea communis Berry Juice

The NMR analyses of all non-polar extracts (A1A, A1B, A2A, and A2B) confirmed the presence of fatty acid esters. The olefinic protons (–C$\underline{H}$=C$\underline{H}$–) of the unsaturated fatty esters appeared as multiplets at $\delta_H$ ca.5.34. The methyl group of the methyl esters (–OCOC$\underline{H}_3$) appeared as singlet at $\delta_H$ 3.65, while the terminal methyl group of the alkyl chain appeared as triplets at $\delta_H$ ca. 0.88 or 0.95, depending on the degree of unsaturation. Moreover, the terminal methylene group of ethyl esters (–OCOC$\underline{H}_2$CH$_3$) appeared as quadruplets at $\delta_H$ ca. 4.12 ($J \approx 6.9$). The vicinal methylene of the esters (–C$\underline{H}_2$COOR) resonated at $\delta_H$ ca. 2.33 (t, $J \approx 6.8$), while the methylene (–C$\underline{H}_2$–) between the double bonds of the unsaturated fatty esters resonated at $\delta_H$ ca. 2.76 (m). The intense signal at $\delta_H$ ca. 1.24 was assigned to the rest of the methylenes of the alkyl chains, partially overlapping the triplet of the terminal methyl group (–OCOCH$_2$C$\underline{H}_3$) of the ethyl esters (Figure 1). Similarly, corresponding signals for the ketones are depicted in Figure 2. In the case of triacylglycerol esters, the peak of the proton at C-2 of glycerol appeared at $\delta_H$ ca. 5.30 as a triplet of triplets, while the methylenes of positions C-1 and C-3 of glycerol resonated at $\delta_H$ ca. 4.33 (q) and 4.17 (q) (Figure 2).

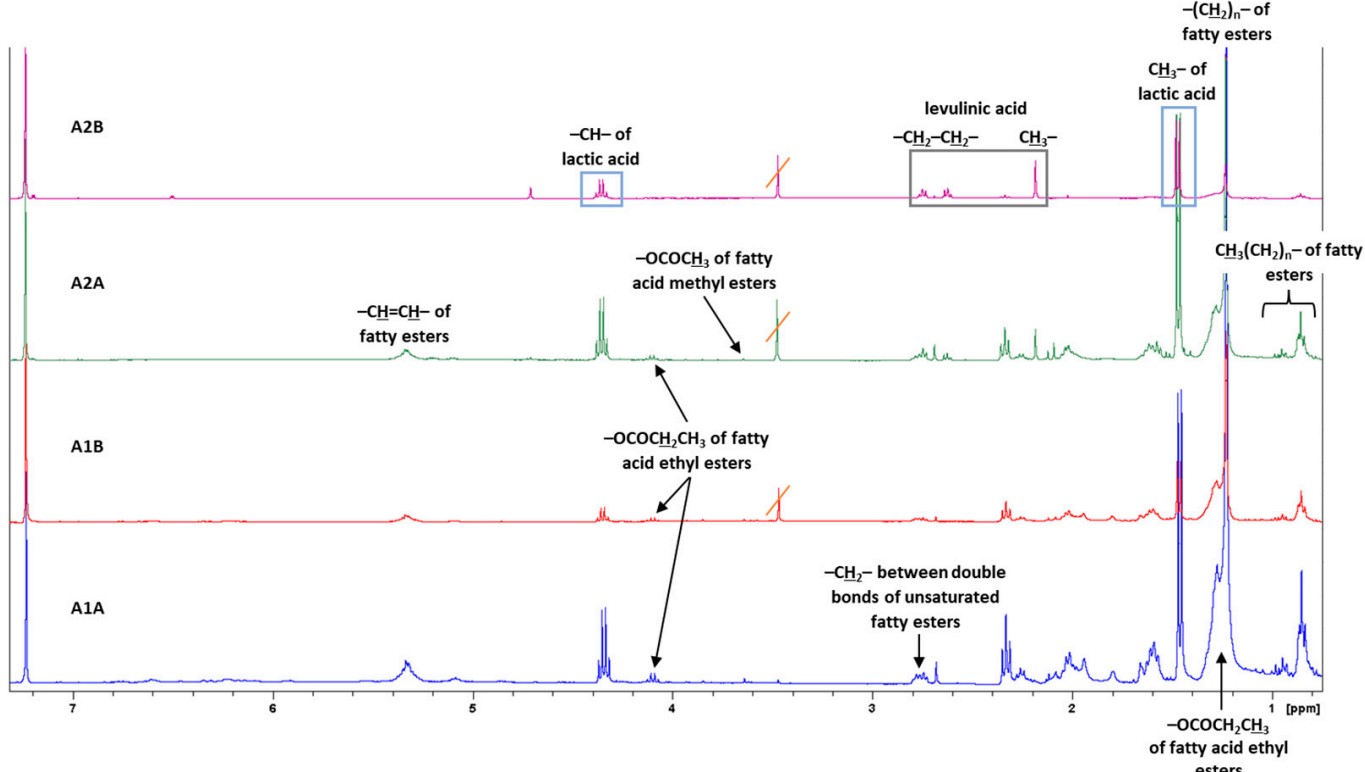

**Figure 1.** $^1$H-NMR spectra (CDCl$_3$, 400 MHz) of non-polar extracts from *Dioscorea communis*. A1A = ether extract; A1B = dichloromethane extract; A2A = ether extract after acid hydrolysis; and A2B = dichloromethane extract after acid hydrolysis.

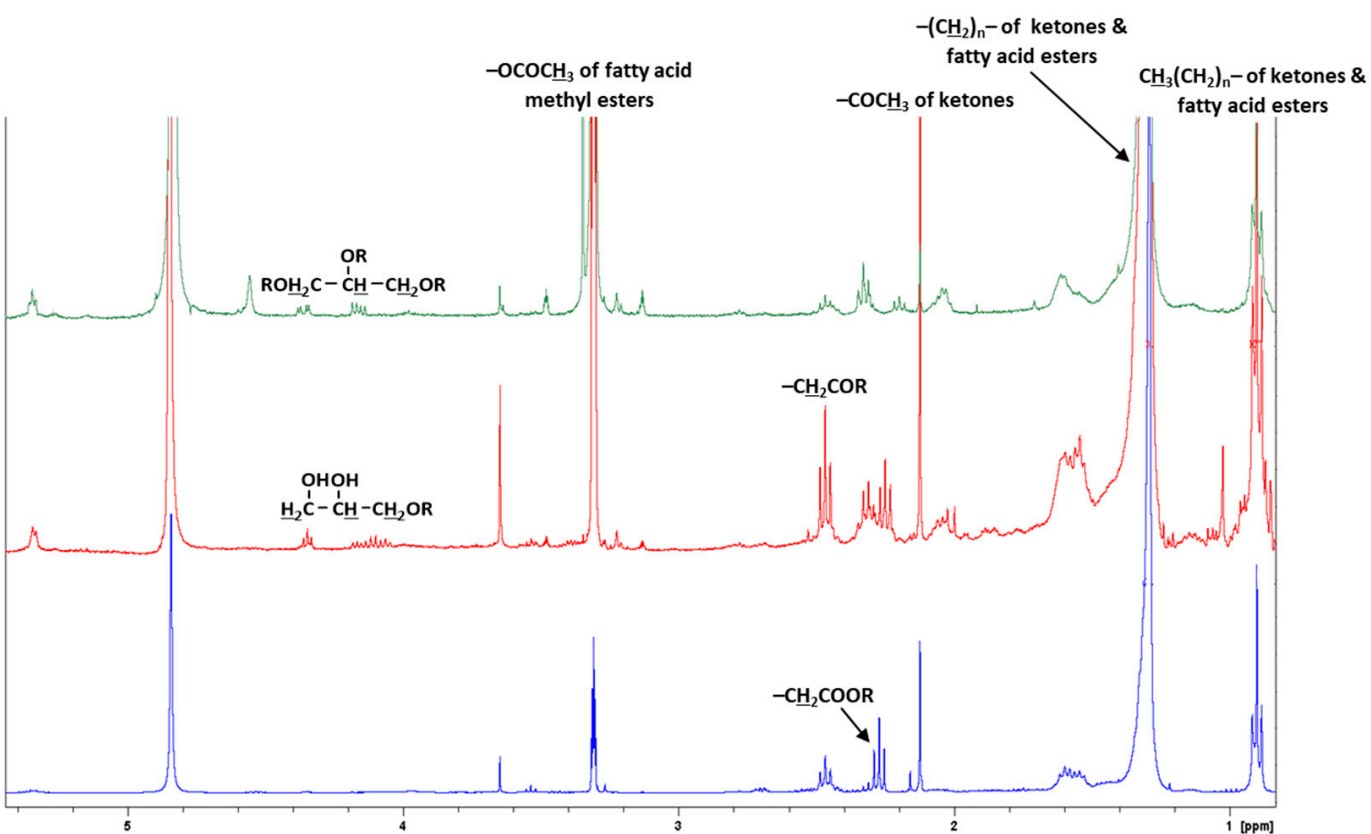

**Figure 2.** $^1$H-NMR spectra (CD$_3$OD, 400 MHz) of the fractions AE, AG, and AI.

In agreement with the GC-MS analyses, both Et$_2$O extracts (A1A and A2A) were mainly characterized by the presence of ethyl esters. In addition, the Et$_2$O extract after acid hydrolysis (A2A) revealed the presence of methyl esters, while the CH$_2$Cl$_2$ extract (A2B) consisted of methyl and ethyl esters. It is worth mentioning that the CH$_2$Cl$_2$ extract after acid hydrolysis (A2B) was remarkably different, since the fatty acid esters and unsaturated derivatives were minor metabolites. The main compounds of this extract were lactic (**1**) and levulinic acids (**2**) not detected through the GC-MS analyses. Concerning the polar *n*-butanol extract (A3), the main signals of the $^1$H-NMR spectrum were assigned to lactic acid (Figure S33).

### 3.3. LC-MS/MS Analysis of n-Butanol Extract of Dioscorea communis Berry Juice

The *n*-butanol extract (A3) of the berry juice obtained after liquid-liquid extraction (Section 2.3) was submitted to LC-MS/MS analysis. The putative identification of these compounds is summarized in Table 8, where the compounds are listed according to their retention times in the total ion chromatogram (TIC) (Figures S34 and S37). Its $^1$H-NMR spectrum was also measured (Figure S33). Based on these results, the main constituent of A3 was lactic acid. Moreover, more than 45 compounds were tentatively identified by LC-MS analysis, including amino acids, organic acids, sugars, fatty acid derivatives, *N*-containing derivatives, flavonoids, phenolic acids, and other phenolic derivatives. The molecular formulas were established based on high-precision quasi-molecular ions such as [M − H]$^-$, [M + CH$_3$COO]$^-$, [M + HCOO]$^-$, [M + H]$^+$, or [M + Na]$^+$ with a mass error of 5.0 ppm, and all information was interpreted and compared with the spectra available in the literature. More specifically, the LC-MS results revealed that the berry juice extract contained various carbohydrates, including mono- and di-saccharides (C$_6$H$_{12}$O$_6$ 203.0530 [M + Na]$^+$, *m/z*, C$_{12}$H$_{22}$O$_{10}$ 349.1121 [M + Na]$^+$, and C$_{12}$H$_{22}$O$_{11}$ 365.1059 [M + Na]$^+$), amino acids like phenylalanine (C$_9$H$_{11}$NO$_2$ 166.0866 [M + H]$^+$), simple organic acids such as ascorbic acid, fumaric acid (C$_6$H$_8$O$_6$ 175.0250 [M − H]$^-$ and C$_4$H$_4$O$_4$ 117.0182 [M + H]$^+$), a variety of

phenolic derivatives including caffeic and cichoric acids, kaempferol glycosides, quercetin and its glycosides (e.g., $C_{27}H_{30}O_{15}$ 539.1516 $[M - H]^-$ and $C_{15}H_{10}O_7$ 301.0353 $[M - H]^-$), as well as tris-(2,4-di-*tert*-butylphenyl)phosphate ($C_{42}H_{63}O_4P$ 685.4366 $[M + Na]^+$), with numerous fatty acid derivatives (Table 8). The chemical evaluation was in agreement with previous reports on the genus *Dioscorea* and the Dioscoreaceae family.

It is noteworthy that the accumulation of fatty acid derivatives and phytosterols is essential during fruit development and ripening. For example, stearic acid, although more abundant in animals, can also be found in vegetable fat. Linolenic acid is mostly found in seeds and berries, while ethyl palmitate is among the most common saturated fatty acid esters in plants [41]. Other commonly detected metabolites in plant extracts, like phytosterols ($\beta$-sitosterol) and triterpenes (amyrin), also have physiological roles in plants. For example, phytosterols are naturally present in plant cell membranes, and triterpenes are associated with plant defense [42]. Furthermore, *N*-alkylamides are essential for plant immunity, usually being produced as a response to abiotic (non-pathogen-induced) and biotic (pathogen-induced) stress. Such compounds act as a chemical defense against phytopathogens and herbivorous predators. Many pathways lead to the expression of defense-related genes, including the production of anti-microbial secondary metabolites like alkylamides [43]. The monitoring of amides in LC-MS was found to be more effectively performed in a positive mode where the carboxamide group is protonated. However, both positive and negative ionization modes were used in the current study, as the negative mode was reported to be more sensitive in the analysis of phenolics and other compounds [44].

### 3.4. Isolated Compounds of Dioscorea communis Berry Juice

Finally, the lyophilized berry juice yielded (Section 2.3) lactic acid (**1**) [45], levulinic acid (**2**) [46], 2-octadecanone (**3**) [47], and two phenolic compounds: the rare tris-(2,4-di-*tert*-butylphenyl)phosphate (**4**), previously isolated from *Vitex negundo* [48], as well as cichoric acid (**5**) [49].

### 3.5. Antibacterial Activity of Dioscorea communis Berry Juice and Selected Fractions

To the best of our knowledge, *D. communis* berry juice has been assessed regarding its antibacterial activity for the first time. Thus far, previous studies on *D. pentaphylla* and *D. bulbifera* extracts and fractions from different plant parts revealed their antibacterial activity [21,50,51].

In the present study, *D. communis* berry juice exhibited bactericidal activity against MRSA and *C. acnes*. Its MIC and MBC values were determined to be 1.56% *w/v* against both bacteria. These results indicate that berry juice might be considered a novel source of antibacterial substances against these two bacteria, which are often implicated in dermatological infections and acne. Moreover, the fraction AD exhibited bacteriostatic activity against *C. acnes*, with an MIC at 6.6 mg/mL. Based on our chemical analyses, the effect could be attributed to 2-octadecanone (compound **3**), methyl stearate, and tris-(2,4-di-tert-butylphenyl)phosphate (compound **4**), which were identified in fraction AD (Figures S12–S18 and Table S1).

### 4. Conclusions

In this study, GC-MS analysis offered influence measurements on the non- and less-polar components with a key role in the characterization of 22 fatty acid derivatives. On the other hand, LC-MS analysis comprises a wide variety of compounds predominant as primary or secondary metabolites, such as amino acids (1), organic acids (3), lipids (14), terpenes-sterols (6), sugars, and phenolics (8), and NMR offers the structure elucidation of 5 individual components, as well as the metabolite fingerprinting. These methods were equally adapted in order to provide both an inclusive impression and complete analysis of the critical components existing in the plant material. The antibacterial activity of *D. communis* berry juice against pathogens often implicated in dermatological infections has been reported herein for the first time. MRSA and *C. acnes* were used, showing MIC

and MBC values at 1.56% *w/v* against both bacteria, which warrants further investigation as this may lead to medical applications. It is notable that these bacteria are resistant to several antibiotics, and treatments that target multiple pathological processes of skin abnormalities are accompanied by side effects [52,53]. Therefore, alternative therapies are urgently needed. Nevertheless, future studies for the evaluation of the acute and sub-acute toxicity effects of the berry juice extract should be conducted.

**Supplementary Materials:** The following supporting information can be downloaded at https://www.mdpi.com/article/10.3390/sci4020021/s1. Figures S1–S40: GC-MS chromatograms of the fractions A1A, A1B, A2A, A2B, AE, AG, AI, $AD_B$, and $AF_A$; Spectra of the known compounds **1**–**5**; LC-MS chromatograms of n-butanol extract (A3) in positive and negative ion mode; LC-MS data of selected compounds in positive and negative ion mode; Flow chart of the isolation procedures; GC-MS tables of $AD_B$ and $AF_A$. Table S1: Chemical composition of fraction $AD_B$. Table S2: Chemical composition of fraction $AF_A$.

**Author Contributions:** Conceptualization, H.S. and M.C.R.; investigation, K.T., C.B., G.L., N.A.D. and M.-E.G.; writing—preparation. C.B., K.T. and N.A.D.; writing—review and editing, H.S., M.C.R. and D.M.; supervision, H.S., M.C.R., D.M. and J.H. All authors have read and agreed to the published version of the manuscript.

**Funding:** This research received no external funding.

**Institutional Review Board Statement:** Not applicable.

**Informed Consent Statement:** Not applicable.

**Acknowledgments:** The authors wish to express their gratitude to Josef Kiermaier and Wolfgang Söllner for recording the MS data (Zentrale Analytik, Faculty of Chemistry and Pharmacy, University of Regensburg). The authors would also like to thank Panagiotis Zotalis for providing us the plant material and relevant information on traditional uses.

**Conflicts of Interest:** The authors declare no conflict of interest.

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
