# Peer review of "Chemical Profile and In Vitro Evaluation of the Antibacterial Activity of Dioscorea communis Berry Juice"

_sci, doi:10.3390/sci4020021_

Round 1

Reviewer 1 Report

Dear Editor in Chief,

Many thanks for letting me review the current draft titled “Chemical profile and in vitro evaluation of the antibacterial activity of Dioscorea communis berry juice” by Tsamia et al. In the current effort, the authors explore the chemical profile of berry juice and assessed the antibacterial effect of this compound. Generally, I suggest accepting the paper after the Major edition by authors.

  • The overall English and sentence formation need improvement.

Abstract:

  • The abstract is too general and didn’t reflect the results which were obtained by the current investigation. Please specify your findings.

  • Authors mentioned that “The results of study provide important information on the chemical characterization of the D. communis berry juice, aiming to unveil its principal metabolites, to further understand its specific antibacterial activity and its occasional toxicity”, so what is this important information? Please explain with the actual results.

Material and methods

  • Please cite the references of the protocols in the material and method section.

Results and discussion

  • The results and discussion section should be organized better. I suggest authors clearly make subtitles for this section and present the results of each section separately.
  • Lines 256-267: In this section, authors started the topic by discussion, while first, they must present their results and then elaborate it with the discussion.
  • Lines 268-271: This paragraph is a duplication of the material and method section and no need to cite in results.
  • The discussion section should be more solid, especially the antibacterial section. Please discuss more and cite more references.
  • For the tables, I believe that drawing the chemical structure of the compound is the more suitable rather molecular formula. Drawing the chemical structure can also increase the visibility of the paper.
  • Lines 400-409. In this paragraph authors talked about the antibacterial effects of the extract, however, at the end of the paragraph authors suddenly mentioned the component of the extract. I believe that authors should relocate the last sentence “Based to our chemical analyses, the main constit- 407 uents of the fraction AD were 2-octadecanone (compound 3), methyl stearate and tris-(2,4- 408 di-tert-butylphenyl)phosphate (compound 4) (Fig. S12-S18; Table S1)” to other parts or should link it with antibacterial concepts using the relevant literature.
  • What is the LD50 of the compound?

Conclusion:

  • The conclusion section doesn’t follow the title and hypothesis of the draft. In the conclusion, the authors have focused on the application of GC–MS analysis and LC–MS analysis, where’s they should talk about their findings. I suggest authors re-write the conclusion and talk about their actual discoveries.

Best wishes

Author Response

We would like to thank the Editor in chief, Editor, and the reviewers for carefully reviewing the manuscript. We much appreciate your comments and suggestions. We have revised the manuscript accordingly. Please find the revised manuscript entitled “Chemical profile and in vitro evaluation of the antibacterial activity of Dioscorea communis berry juice” for publication in Sci. The comments have been addressed accordingly and highlighted (with red color) in the manuscript. The reviewers’ comments are in plain text and the response are marked in blue here below.

Linguistic and grammatical errors have been checked and revised thoroughly in the manuscript we hope it now matches the journal standard.

Reviewer: 1

In the current effort, the authors explore the chemical profile of berry juice and assessed the antibacterial effect of this compound. Generally, I suggest accepting the paper after the Major edition by authors. The overall English and sentence formation need improvement.

Answer

The manuscript has been revised according to the reviewer's suggestion. The manuscript has been re-written/re-organised, and mistakes in grammar and content were corrected.

Abstract:

The abstract is too general and didn’t reflect the results which were obtained by the current investigation. Please specify your findings.

 Authors mentioned that “The results of the study provide important information on the chemical characterization of the D. communis berry juice, aiming to unveil its principal metabolites, to further understand its specific antibacterial activity and its occasional toxicity”, so what is this important information? Please explain with the actual results.

Answer

Thank you for this comment, we agree, and the abstract has been changed as follows:

 ‘’This work revealed the presence of several metabolites belonging to different phytochemical groups, such as fatty acid esters, alkylamides, phenolic derivatives, and organic acids with lactic acid being predominant. In parallel, based on orally transmitted traditional uses, the initial extract and selected fractions were tested in vitro for their anti-bacterial effects and exhibited good activity against two bacterial strains related to skin infections, i.e. methicillin-resistant Staphylococcus aureus and Cutibacterium acnes. The MIC and MBC values of the extract were determined at 1.56 % w/v against both bacteria. The results of this study provide important information on the chemical characterization of the D. communis berry juice, unveiling the presence of 71 metabolites, that might contribute and further explain its specific antibacterial activity and its occasional toxicity.’’

Material and methods

Please cite the references of the protocols in the material and method section.

Answer

Thank you for this comment. We corrected and further superscripted the corresponding reference citations as suggested.

‘’[23] Grafakou, M. E.; Barda, C.; Heilmann, J.; Skaltsa, H. Macrocyclic Diterpenoid Constituents of Euphorbia deflexa, an Endemic Spurge from Greece. J Nat Prod 2021, 84 (11), 2893-2903; DOI:10.1021/acs.jnatprod.1c00654.

[24] Grafakou, M. E.; Diamanti, A.; Simirioti, E.; Terezaki, A.; Barda, C.; Sfiniadakis, I.; Rallis, M.; Skaltsa, H. Wound Healing Effects from 3 Hypericum spp. Essential Oils. Planta Medica Int Open 2021, 8 (02), e69-e77; DOI:10.1055/a-1492-3634.

[27] Tremmel, M.; Paetz, C.; Heilmann, J. In Vitro Liver Metabolism of Six Flavonoid C-Glycosides. Molecules 2021, 26, 6632; DOI:10.3390/molecules26216632.’’

Results and discussion

The results and discussion section should be organized better. I suggest authors clearly make subtitles for this section and present the results of each section separately.

Answer

Thank you for your comment. We have made the necessary corrections. In the revised version subtitles in the section ‘’results and discussion’’ are included.

‘’3.1 Chemical composition by GC-MS analyses in various fractions of Dioscorea communis berry juice.

3.2 NMR analyses in non-polar fractions of Dioscorea communis berry juice

3.3 LC-MS/MS analysis of n-butanol extract of Dioscorea communis berry juice

3.4 Isolated compounds of Dioscorea communis berry juice

3.5 Antibacterial activity of Dioscorea communis berry juice and selected fractions’’

Lines 256-267: In this section, authors started the topic by discussion, while first, they must present their results and then elaborate it with the discussion.

Answer

We briefly summarized the results in the beginning of the section Results and Discussion, while we extensively analyze them in the following subchapters. This enables the reader to better follow the evaluation of the results.

Lines 268-271: This paragraph is a duplication of the material and method section and no need to cite in results. The discussion section should be more solid, especially the antibacterial section. Please discuss more and cite more references.

Answer

These four lines are just a small introductory section. If the reviewer agrees we prefer to keep it.

For the tables, I believe that drawing the chemical structure of the compound is the more suitable rather molecular formula. Drawing the chemical structure can also increase the visibility of the paper.

Answer

We can understand the concern raised by the reviewer. Indeed, drawing the chemical structure can also increase the visibility of the paper, though based on the nature of the molecules the chemical structures are not comprehensible as they consist long-chain fatty acids. In addition, the reference to the number of carbons in such molecules is essential e.g. C18 easily gives an account of the octadecane- or stearic backbone.

Lines 400-409. In this paragraph authors talked about the antibacterial effects of the extract, however, at the end of the paragraph authors suddenly mentioned the component of the extract. I believe that authors should relocate the last sentence “Based to our chemical analyses, the main constituents of the fraction AD were 2-octadecanone (compound 3), methyl stearate and tris-(2,4-di-tert-butylphenyl)phosphate (compound 4) (Fig. S12-S18; Table S1)” to other parts or should link it with antibacterial concepts using the relevant literature.

What is the LD50 of the compound?

Answer

We have made the necessary corrections according to the reviewer's suggestions.

Lethal dose of the compounds has not been performed. Indeed, it would be a nice suggestion for future studies.

Conclusion:

The conclusion section doesn’t follow the title and hypothesis of the draft. In the conclusion, the authors have focused on the application of GC–MS analysis and LC–MS analysis, where’s they should talk about their findings. I suggest authors re-write the conclusion and talk about their actual discoveries.

Answer

We have made the necessary corrections according to the reviewer's suggestions. Actual discoveries have been included.

As follows:

’In this study, GC–MS analysis offers influence measurements on the non and less polar components with a key role in the characterization of 22 fatty acid derivatives. On the other hand, LC-MS covers a large array of compounds predominant as primary or secondary metabolites such as amino acid (1), organic acids (3) lipids (14), terpenes-sterols (6) sugars, and phenolics (8), and NMR offers the structure elucidation of 5 individual components, as well as the metabolite fingerprinting. These methods were equally adapted to allow both inclusive impression and full analysis of critical components of the plant material. The antibacterial activity of D. communis berry juice against pathogens often implicated in dermatological infections has been reported, herein, for the first time. MRSA and C. acnes were used, showing MIC and MBC values at 1.56 % w/v against both bacteria, which warrants further investigation as this may lead to medical applications. It is notable that these bacteria are resistant to several antibiotics and treatments that target multiple pathological processes of skin abnormalities are accompanied by side effects [52, 53]. Therefore, alternative therapies are urgently needed. Nevertheless, future studies for the evaluation of the acute and sub-acute toxicity effects of the berry juice extract should be conducted.’’

Reviewer 2 Report

This manuscript reported phytochemical research on the fruit juice of Dioscorea communis using various approaches, including chromatographic methods (column chromatography, LC–MS, and GC–MS) and direct spectroscopic methods using NMR. The authors reported the presence of several 20 metabolites belonging to different phytochemical groups, such as fatty acid esters, alkylamides, 21 phenolic derivatives, and organic acids with lactic acid being predominant.

I read with interest the manuscript. This manuscript will be suitable for publishing once the minor changes indicated below are made.

Please correct some mistakes and typos.

Lane 40-41 :  D. pyrenaica >> Dioscorea pyrenaica, D. chouardii >> Dioscorea chouardii, D. communis >> Dioscorea communis, (The scientific name used for the first time in the text should be expressed as the full name.)

Lane 70, 72-73 : E. coli, E. aerogenes, K. pneumoniae, M. smegmatis and M. tuberculosis, >> full name (The scientific name used for the first time in the text should be expressed as the full name.)

Lane 42 : genus (italic) >> genus (normal)

Lane 45 : D. (normal) >> D. (italic)

Lane 63 : in vitro in vivo (normal) >> in vitro in vivo. (italic)

Lane 69 : et al. (normal) >> et al. (italic)

Lane 92 : 1H-NMR 600.25 MHz, 13C-NMR 150.95 MHz >> 1H-NMR 600 MHz, 13C-NMR 150 MHz

Lane 95 : COSY (COrrelation SpectroscopΥ) >> Y : Change the font of Y to "Times New Roman".

Lnae 96-97 : delete NOESY >> There is no NOESY result in the result and supplementary information.

Lane 98 and 165 : The q-TOF equipment name is different between Lane 98 (6540) and Lane 165 (G6540A).

Lane 105, 138 : x >> ´ (As described in lane 166, use symbol to indicate multiplication.)

Lane 156 : n-alkanes >> n-alkanes

Lane 185 : Cutibacterium acnes >> Cutibacterium acnes (When the sub-title is written as italic, the scientific name is displayed as normal.)

Lane 209 : methicillin resistant Staphylococcus aureus (MRSA) strain 1552 and Cutibacterium acnes >> MRSA strain and C. acnes (Methicillin resistant Staphylococcus aureus (MRSA) is already expressed in lanes 78-79.)

Lane 244 : - >> - (use symbol to indicate subtraction)

Lane 262 : O- and C- glucoside >> O- and C- glucoside (use italic)

Lane 292, 294, 296, 299, 345, : D. communis >> Dioscorea communis (In table or figure legnds, use the scientific name as the full name.)

Lane 299 : Table 1. >> Table 4.

Lane 310, 356, 379 : N- and P- containing >> N- and P- containing (use italic)

Lane 319 : C9-C25 n-alkanes>> C9-C25 n-alkanes

Lane 333 : ca. 4.33 and 4.17m, both as doublets of doublets >> ca. 4.33 (q) and 4.17 (q)

Lane 352 : n-butanol >> n-butanol

In Figure 1, CH2 of lactic acid >> CH3 of lactic acid

In Figure 1, CH3(CH2)n of fatty esters >> CH3(CH2)n of fatty esters

In Table 1-5, delete expressions of “all-cis” or “trans” and use italic for α, β, R, Z, or E, etc

   ex1. cis-9 >> 9Z

   ex2. cis-9,12 >> 9Z,12Z-

   ex3. all-cis-9,12,15 >> 9Z,12Z,15Z -

   ex4, trans-9 >> 9E (like ex.1~3)

Lane 367, 394, 409 : tris-(2,4-di-tert-butylphenyl)phos- >> tris-(2,4-di-tert-butylphenyl)phos-

Author Response

We would like to thank the Editor in chief, Editor, and the reviewers for carefully reviewing the manuscript. We much appreciate your comments and suggestions. We have revised the manuscript accordingly. Please find the revised manuscript entitled “Chemical profile and in vitro evaluation of the antibacterial activity of Dioscorea communis berry juice” for publication in Sci. The comments have been addressed accordingly and highlighted (with red color) in the manuscript. The reviewers’ comments are in plain text and the response are marked in blue here below.

Linguistic and grammatical errors have been checked and revised thoroughly in the manuscript we hope it now matches the journal standard.

Reviewer: 2

This manuscript reported phytochemical research on the fruit juice of Dioscorea communis using various approaches, including chromatographic methods (column chromatography, LC–MS, and GC–MS) and direct spectroscopic methods using NMR. The authors reported the presence of several 20 metabolites belonging to different phytochemical groups, such as fatty acid esters, alkylamides, 21 phenolic derivatives, and organic acids with lactic acid being predominant.

I read with interest the manuscript. This manuscript will be suitable for publishing once the minor changes indicated below are made.

Answer:

We would like to thank the reviewer for his/her valuable comments.

Please correct some mistakes and typos.

Lane 40-41 :  D. pyrenaica >> Dioscorea pyrenaica, D. chouardii >> Dioscorea chouardii, D. communis >> Dioscorea communis, (The scientific name used for the first time in the text should be expressed as the full name.)

Lane 70, 72-73 : E. coli, E. aerogenes, K. pneumoniae, M. smegmatis and M. tuberculosis, >> full name (The scientific name used for the first time in the text should be expressed as the full name.)

Lane 42 : genus (italic) >> genus (normal)

Lane 45 : D. (normal) >> D. (italic)

Lane 63 : in vitro in vivo (normal) >> in vitro in vivo. (italic)

Lane 69 : et al. (normal) >> et al. (italic)

Lane 92 : 1H-NMR 600.25 MHz, 13C-NMR 150.95 MHz >> 1H-NMR 600 MHz, 13C-NMR 150 MHz

Lane 95 : COSY (COrrelation SpectroscopΥ) >> Y : Change the font of Y to "Times New Roman".

Lnae 96-97 : delete NOESY >> There is no NOESY result in the result and supplementary information.

Lane 98 and 165 : The q-TOF equipment name is different between Lane 98 (6540) and Lane 165 (G6540A).

Lane 105, 138 : x >> ´ (As described in lane 166, use symbol to indicate multiplication.)

Lane 156 : n-alkanes >> n-alkanes

Lane 185 : Cutibacterium acnes >> Cutibacterium acnes (When the sub-title is written as italic, the scientific name is displayed as normal.)

Lane 209 : methicillin resistant Staphylococcus aureus (MRSA) strain 1552 and Cutibacterium acnes >> MRSA strain and C. acnes (Methicillin resistant Staphylococcus aureus (MRSA) is already expressed in lanes 78-79.)

Lane 244 : - >> - (use symbol to indicate subtraction)

Lane 262 : O- and C- glucoside >> O- and C- glucoside (use italic)

Lane 292, 294, 296, 299, 345, : D. communis >> Dioscorea communis (In table or figure legnds, use the scientific name as the full name.)

Lane 299 : Table 1. >> Table 4.

Lane 310, 356, 379 : N- and P- containing >> N- and P- containing (use italic)

Lane 319 : C9-C25 n-alkanes>> C9-C25 n-alkanes

Lane 333 : ca. 4.33 and 4.17m, both as doublets of doublets >> ca. 4.33 (q) and 4.17 (q)

Lane 352 : n-butanol >> n-butanol

In Figure 1, CH2 of lactic acid >> CH3 of lactic acid

In Figure 1, CH3(CH2)n of fatty esters >> CH3(CH2)n of fatty esters

In Table 1-5, delete expressions of “all-cis” or “trans” and use italic for α, β, R, Z, or E, etc

   ex1. cis-9 >> 9Z

   ex2. cis-9,12 >> 9Z,12Z-

   ex3. all-cis-9,12,15 >> 9Z,12Z,15Z -

   ex4, trans-9 >> 9E (like ex.1~3)

Lane 367, 394, 409 : tris-(2,4-di-tert-butylphenyl)phos- >> tris-(2,4-di-tert-butylphenyl)phos-

Answer:

We thank the reviewer for the time spent to our manuscript. All corrections have been done point by point.

Linguistic and grammatical errors have been checked and revised thoroughly in the manuscript. We hope it now matches the journal standard.

Reviewer 3 Report

Dear authors, 

 first of all let me congratulate you on your hard work and interesting research topic. Secondly, I would like to make some observations and try to give constructive feedback on your work. 

Generally the article is well written and structured, however there are some issues with the analytical part of your manuscript:

1) Some of the techniques mentioned in the Materials and methods are newer mentioned again in the experimental part (for example UV spectrophotometry - no mentioned is made of UV spectra, how they were used to identify any of the compounds)

2) The GC-MS chromatograms, as per the method described, should be 90 minutes long. I do understand that not all regions of a chromatogram are of interest, but the chromatograms presented vary so much in which areas they present, that it almost seems cherry-picked. I would recommend showing either the entire chromatograms (at least for some representative samples) and the presenting zoomed in regions of the same chromatograms for the areas with the compounds identified; or showing a fixed area for all chromatograms presented (e.g. 30-80 minutes, or whichever is relevant) where all the compounds elute even if they are not present in all samples. Otherwhise it seems that you only present small portion and are omitting parts which might contain useful data.

3) No mass spetra are presented in the manuscript for neither GC-MS or LC-MS analysis, which would support identification of compounds. It might be too much to ask for all the mass spectra, but a few representative might spectra would be helpful to support the conclusions.

4) For LC-MS analysis there is only a TIC chromatogram, without any additional information (extracted chromatograms, mass spectra etc.), which is . 

5) Please discuss what the relevance of comparing a negative and positive ionization TIC is, why this is helpful? Further explanations for these chromatograms would also be needed to make sence of the results presented and obtained through LC-MS analysis (did you do fragmentation for the parent ions, what type of fragmentation, with what collision energy?). Given you have used a highly sensitive and selective mass detector (TOF) there is a lot of information which can be extracted from the LC-MS analysis, which is however is only mentioned but not discussed.

6) Some of the figures are numbered incorrectly (for example Fig S22 is presented in the manuscript text as a TIC obtained from LC-MS analysis, but it is in fact a spectra obtained after H-NMR). Please check and make sure are figures are labeled accordingly.

Author Response

We would like to thank the Editor in chief, Editor, and the reviewers for carefully reviewing the manuscript. We much appreciate your comments and suggestions. We have revised the manuscript accordingly. Please find the revised manuscript entitled “Chemical profile and in vitro evaluation of the antibacterial activity of Dioscorea communis berry juice” for publication in Sci. The comments have been addressed accordingly and highlighted (with red color) in the manuscript. The reviewers’ comments are in plain text and the response are marked in blue here below.

Linguistic and grammatical errors have been checked and revised thoroughly in the manuscript we hope it now matches the journal standard.

Reviewer: 3

Dear authors, first of all let me congratulate you on your hard work and interesting research topic. Secondly, I would like to make some observations and try to give constructive feedback on your work.

Generally the article is well written and structured, however there are some issues with the analytical part of your manuscript:

1) Some of the techniques mentioned in the Materials and methods are newer mentioned again in the experimental part (for example UV spectrophotometry - no mentioned is made of UV spectra, how they were used to identify any of the compounds)

Answer

Thank you for this comment. We have made the necessary corrections. In fact, UV spectrophotometry was mentioned by oversight.

2) The GC-MS chromatograms, as per the method described, should be 90 minutes long. I do understand that not all regions of a chromatogram are of interest, but the chromatograms presented vary so much in which areas they present, that it almost seems cherry-picked. I would recommend showing either the entire chromatograms (at least for some representative samples) and the presenting zoomed in regions of the same chromatograms for the areas with the compounds identified; or showing a fixed area for all chromatograms presented (e.g. 30-80 minutes, or whichever is relevant) where all the compounds elute even if they are not present in all samples. Otherwhise it seems that you only present small portion and are omitting parts which might contain useful data.

Answer

We revised the supporting information according to the reviewer's suggestion. Both chromatograms i.e annotated, and total are included. We understand the concern raised by the reviewer; indeed we present selected chromatogram areas but only for data interpretation purposes.

3) No mass spectra are presented in the manuscript for neither GC-MS or LC-MS analysis, which would support identification of compounds. It might be too much to ask for all the mass spectra, but a few representative might spectra would be helpful to support the conclusions.

Answer

Indeed, the data preparation has been performed due to the large data volumes (e.g. positive mode analysis report 144 pg) we omitted them. In the revised version representative more data are included.

4) For LC-MS analysis there is only a TIC chromatogram, without any additional information (extracted chromatograms, mass spectra etc.), which is .

Answer

All chromatograms are included in the revised version of the supporting information.

5) Please discuss what the relevance of comparing a negative and positive ionization TIC is, why this is helpful?

 Further explanations for these chromatograms would also be needed to make sence of the results presented and obtained through LC-MS analysis (did you do fragmentation for the parent ions, what type of fragmentation, with what collision energy?). Given you have used a highly sensitive and selective mass detector (TOF) there is a lot of information which can be extracted from the LC-MS analysis, which is however is only mentioned but not discussed.

Answer

We would like to thank the reviewer for this comment, we agree that our comment on the comparison of negative and positive ionization is wordy. This comment aims to support the assignment of amides in positive ion mode. Moreover, our general approach in this study included many analytical techniques among them also LC-MS. It is a fact that a plethora of information can be extracted from such analysis which could stand alone as an individual project.  We had no intention to overlap our work with this method. The assignments were supported by the fragmentation patterns and the literature under the guidelines of Zentrale Analytik, Faculty of Chemistry and Pharmacy.

6) Some of the figures are numbered incorrectly (for example Fig S22 is presented in the manuscript text as a TIC obtained from LC-MS analysis, but it is in fact a spectra obtained after H-NMR). Please check and make sure are figures are labeled accordingly.

Answer

Supporting information has been duly revised.

Reviewer 4 Report

1. The figures along the manuscript is not clear, please modify. 
2. Add the best result in the abstract 
3. Compare this method with traditional methods
4. The aim of this research is not clear

Author Response

We would like to thank the Editor in chief, Editor, and the reviewers for carefully reviewing the manuscript. We much appreciate your comments and suggestions. We have revised the manuscript accordingly. Please find the revised manuscript entitled “Chemical profile and in vitro evaluation of the antibacterial activity of Dioscorea communis berry juice” for publication in Sci. The comments have been addressed accordingly and highlighted (with red color) in the manuscript. The reviewers’ comments are in plain text and the response are marked in blue here below.

Linguistic and grammatical errors have been checked and revised thoroughly in the manuscript we hope it now matches the journal standard.

Reviewer: 4

  1. The figures along the manuscript is not clear, please modify them.

Answer

Figures have been improved and revised in the resubmitted manuscript.

  1. Add the best result in the abstract

Answer

The Abstract has been revised.

  1. Compare this method with traditional methods

Answer

The applied research methods in this study are traditionally used for phytochemical research. We propose no further discussion.

  1. The aim of this research is not clear

Answer

The aim of the study is presented in lines 87-91.

Round 2

Reviewer 1 Report

I have no more comments and the authors pointed out all of the comments. I suggest accepting the paper in the current format.

Best of luck

Author Response

We would like to thank the Editor in chief, Editor, and the reviewers. We much appreciate your comments. We have revised the manuscript accordingly. Please find the revised manuscript entitled “Chemical profile and in vitro evaluation of the antibacterial activity of Dioscorea communis berry juice” for publication in Sci. The comments have been addressed accordingly and highlighted (with red color) in the manuscript.

Indeed, a self-plagiarism has been detected, due to the fact that we reused mandatory portions of our methodology (e.g. Primary sources 1-3). There are only a limited number of ways that methods can be explained. All the rest duplicate places with more than 12 continuous words have been revised e.g.:

Results and Discussion:

1% plagiarism detected as following: “Furthermore, N-alkylamides are essential for plant immunity, usually produced as a response to abiotic (non-pathogen-induced) and biotic (pathogen-induced) stress. Many pathways lead to the expression of defense-related genes including the production of anti-microbial secondary metabolites, like alkylamides. Such compounds are secondary metabolites produced in response to stress and act as a chemical defense against plant competitors or microbial and herbivorous predators [43]”, was revised accordingly: “Furthermore, N-alkylamides are essential for plant immunity, usually produced as a response to abiotic and biotic stress. Such compounds act as a chemical defense against phytopathogens and herbivorous predators. Many pathways lead to the expression of defense-related genes including the production of anti-microbial specialized metabolites, like alkylamides [43]”.

Conclusion:

1% plagiarism detected as following: “On the other hand, LC-MS covers a large array of compounds predominant … as well as the metabolite fingerprinting. These methods were equally adapted to allow both inclusive impression and full analysis of critical components of the plant material”, was revised accordingly: “On the other hand, LC-MS analysis comprises a wide variety of compounds predominant … as well as the metabolite fingerprinting. These methods were equally adapted in order to provide both inclusive impression and complete analysis of critical components existing in the plant material”.